# Knockout of PA200 improves proteasomal degradation and myelination in a proteotoxic neuropathy

Jordan JS VerPlank[1] , Joseph M Gawron[1], Nicholas J Silvestri[2], Lawrence Wrabetz[1,2], Maria Laura Feltri[1,2,3,4]

The cellular response to a decrease in protein degradation by 26S proteasomes in chronic diseases is poorly understood. Pharmacological inhibition of proteasomes increases the expression of proteasome subunits and Proteasome Activator 200 (PA200), an alternative proteasome activator. In the S63del mouse model of the peripheral neuropathy Charcot Marie Tooth 1B (CMT1B), proteasomal protein degradation is decreased and proteasome gene expression is increased. Here, we show an increase in PA200 and PA200-bound proteasomes in the peripheral nerves of S63del mice. To test genetically whether the upregulation of PA200 was compensatory, we generated S63del//PA200−/− mice. Unexpectedly, in the sciatic nerves of these mice, there was greater proteasomal protein degradation than in S63del, less polyubiquitinated proteins and markers of the unfolded protein response, and a greater amount of assembled, active 26S proteasomes. These changes were not seen in PA200−/− controls and were therefore specific to the neuropathy. Furthermore, in S63del//PA200−/− mice, myelin thickness and nerve conduction were restored to WT levels. Thus, the upregulation of PA200 is maladaptive in S63del mice and its genetic ablation prevented neuropathy.

## Introduction

In mammalian cells, the great majority of proteins are degraded by the ubiquitin–proteasome system (UPS) (Collins & Goldberg, 2017). In this pathway, proteins are conjugated to ubiquitin and degraded by the 26S proteasome, which is composed of a 20S core particle with a 19S regulatory particle on one or both ends. Polyubiquitinated proteins are first bound by subunits of the 19S, then deubiquitinated, unfolded, and translocated into the hollow central chamber of the 20S where they are hydrolyzed into small peptides (Mao, 2021).

The 26S proteasome's ability to degrade proteins is tightly regulated and one newly appreciated mechanism of regulation is phosphorylation (VerPlank & Goldberg, 2017). The cAMP-dependent kinase (PKA) (Lokireddy et al, 2015; VerPlank et al, 2019) and the cGMP-dependent kinase (PKG) (VerPlank et al, 2020) have been shown to phosphorylate the proteasome at separate sites, stimulating proteasomal activities and the degradation of distinct categories of proteins in cells. These findings are of therapeutic interest because decreases in protein degradation by 26S proteasomes have been reported in vertebrate models of several diseases, especially neurodegenerative diseases that are caused by the expression of mutant, aggregation-prone proteins (Le Guerroué & Youle, 2021). The resulting slow rate of degradation leads to the accumulation in the affected cells of the causative mutant protein and un-degraded, potentially toxic proteins that contribute to disease progression.

Mutations in myelin protein zero (MPZ) cause the hereditary peripheral neuropathy Charcot Marie Tooth 1B (Fridman & Saporta, 2021). MPZ is synthesized in Schwann cells, the myelinating glia of the peripheral nervous system, and constitutes ~50% of the total protein content of peripheral nerves (Siems et al, 2020). Over 200 mutations in MPZ are known to cause hereditary neuropathies (Callegari et al, 2019; Fridman & Saporta, 2021). The deletion of serine 63 (MPZ$^{S63del}$) causes CMT1B through gain of toxic function mechanisms (Wrabetz et al, 2006). In S63del transgenic mice, the MPZ$^{S63del}$ protein accumulates in the ER of Schwann cells (Sidoli et al, 2016) and induces a canonical unfolded protein response (UPR) (Pennuto et al, 2008; D'Antonio et al, 2013). In the peripheral nerves of these mice, the ability of 26S proteasomes to hydrolyze its substrates is decreased, the rate of proteasomal protein degradation is reduced, and consequently, polyubiquitinated proteins accumulate (VerPlank et al, 2018). Treating S63del mice for 2 wk with the PDE5 inhibitor sildenafil, to raise cGMP and activate PKG, increased the 26S proteasome's enzymatic activities, reduced the UPR and the accumulation of polyubiquitinated proteins, and restored myelin thickness and nerve conduction (VerPlank et al, 2022). Thus, activating the proteasome with agents that promote its phosphorylation by PKG

[1]Department of Biochemistry, Institute for Myelin and Glia Exploration, Jacobs School of Medicine and Biomedical Sciences, State University of New York at Buffalo, Buffalo, NY, USA [2]Department of Neurology, Institute for Myelin and Glia Exploration, Jacobs School of Medicine and Biomedical Sciences, State University of New York at Buffalo, Buffalo, NY, USA [3]IRCCS Neurological Institute "Carlo Besta," Milano, Italy [4]Department of Medical Biotechnology and Translational Medicine, Universita' degli Studi di Milano, Milano, Italy

Correspondence: jordan.verplank@usuhs.edu
Jordan JS VerPlank's present address is Department of Anatomy, Physiology, and Genetics, F. Edward Hebert School of Medicine, Uniformed Services University of the Health Sciences, Bethesda, MD, USA

represents a promising new approach to treat CMT1B, and possibly other diseases in which protein degradation is impaired.

Although decreases in protein degradation have now been reported in models of many diseases and aging (Hipp et al, 2019), the cellular response to proteasome impairment in chronic diseases has received little attention. The majority of prior studies used proteasome inhibitors, which are therapies for multiple myeloma (Goldberg, 2012). In response to pharmacological proteasome inhibition, cells upregulate all proteasome subunits via the transcription factor Nrf1 (Sha & Goldberg, 2014), and autophagy genes (Sha et al, 2018a), presumably in an attempt to maintain proteostasis. In the peripheral nerves of S63del mice, Nrf1 translocates to the nucleus and proteasome subunits are upregulated (VerPlank et al, 2018). Proteasome inhibitors also cause the upregulation of Proteasome Activator 200 (PA200) (Sha & Goldberg, 2014) and an increase in PA200-bound proteasomes (Welk et al, 2016).

PA200 is one of four alternative activators of the proteasome. The other three, PA28α, β, and ɣ, are not upregulated in response to proteasome inhibition (Sha & Goldberg, 2014). All four bind the ends of the 20S and open the gate, permitting access to the proteasome's proteolytic active sites. The alternative activators, unlike the 19S, do not bind, unfold and translocate polyubiquitinated proteins (Wang et al, 2020). Therefore, their roles in intracellular protein degradation by proteasomes are unclear. PA200 is enriched in the testes, where it is necessary for spermatogenesis (Khor et al, 2006; Huang et al, 2016). Its functions in other tissues have received little attention, but PA200−/− mice have no reported defects other than infertility in males (Khor et al, 2006). PA200 has been shown to participate in the proteasomal degradation of specific proteins (e.g., acetylated histones) (Qian et al, 2013; Mandemaker et al, 2018), but its involvement in protein degradation generally, and its role in the response to proteasome inhibition, are unknown.

Here, we show that PA200 mRNA, protein, and -associated proteasomes are increased in the sciatic nerves of S63del mice. To better understand whether this increase in PA200 was involved in the compensatory response to maintain protein degradation when 26S proteasomes are impaired, we generated S63del//PA200−/− mice and evaluated in their peripheral nerves the rates of protein degradation and proteasome peptide hydrolysis. Surprisingly, S63del//PA200−/− mice exhibited increased rates of protein degradation and proteasomal activity than S63del mice, prompting us to identify a possible mechanism for this unexpected, disease-specific finding. Because increasing proteasomal degradation and peptidase activity in S63del with pharmacological agents that raised cGMP had therapeutic effects, we also evaluated whether the S63del//PA200−/− mice manifested the proteotoxic stress, hypomyelination and slow nerve conduction that is seen in S63del mice.

# Results

## PA200 is upregulated in the sciatic nerves of S63del mice

The levels of mRNA of PSME4, the gene that encodes the protein PA200, are increased by ~50% in the sciatic nerves of S63del mice (Fig 1A). This magnitude of increase is comparable to that observed previously in transcriptomic studies on sciatic nerves of WT and S63del mice (D'Antonio et al, 2013). The levels of PA200 protein are ~fourfold greater in S63del than in WT, as seen by Western blot analysis of sciatic nerve lysates (Fig 1B).

PA200 contains a bipartite nuclear localization sequence (NLS) and was detected as a nuclear protein in HeLa cells in the first report that described PA200 as an activator of the 20S proteasome (Ustrell et al, 2002). To evaluate PA200's localization in sciatic nerves, we performed subcellular fractionation in buffer conditions in which 26S proteasomes remain assembled and functional (VerPlank et al, 2020). Unexpectedly, PA200 was located predominantly in the cytosolic fraction of sciatic nerves of both WT and S63del mice (Fig 1C). Schwann cells constitute ~90% of the dry mass of sciatic nerves (Rutkowski et al, 1992). Therefore, PA200 is located predominantly in the cytosolic fraction of Schwann cells. In sciatic nerves from S63del mice, PA200 protein was increased in both the cytosolic and nuclear fractions, but most noticeably in the cytosolic fraction (Fig 1C).

Lysates of sciatic nerves from PA200−/− mice (Khor et al, 2006) were included as controls in the above experiments. PCR analysis of genomic DNA from these mice showed biallelic deletion of PSME4 (Fig S1) and, accordingly, PSME4 mRNA was undetectable via qRT-PCR in the sciatic nerves of PA200−/− mice (Fig 1A). In WT and S63del mice, Western blot analysis with antibodies against PA200 detected two bands: one with the expected molecular weight of ~200 kD and another with a molecular weight of ~150 kD (Fig 1B and C). In PA200−/− lysates, the 200 kD band was absent, whereas the 150 kD band was present. Notably, this 150 kD band was detected more readily than the 200 kD band (Fig 1B and C) and was located primarily in the nuclear fraction of WT and S63del sciatic nerves (Fig 1C). This nonspecific band detected by some commercially available antibodies (Welk et al, 2019) may be the reason PA200 has been repeatedly reported to be an exclusively nuclear protein.

## PA200-bound proteasomes are increased in the sciatic nerves of S63del mice

To test whether the increase in the levels of PSME4 mRNA and PA200 protein led to an increase in PA200-bound proteasomes, we first immunoprecipitated proteasomes from the sciatic nerve lysates of WT and S63del mice with an antibody against the 20S proteasome subunit α1 and then immunoblotted for PA200. In S63del, more PA200 protein was detected in the proteasome immunoprecipitate than in WT (Fig 2A).

Centrifugation of the sciatic nerve lysates through a 10–50% glycerol gradient followed by measurements of proteasomal peptidase activity and Western blot analysis showed that PA200 protein is found only in fractions that contained the 20S proteasome subunit α1 and proteasomal chymotrypsin-like activity in both WT (fractions 1–4) and S63del (fractions 1–7) (Fig 2B). PA200 protein was not found in fractions that did not contain a 20S proteasome subunit or peptidase activity (fractions 10–19), and little to no PA200 was found in the unbound fraction of the proteasome immunoprecipitate (Fig 2A), strongly suggesting that the majority of PA200 protein in the sciatic nerves of WT and S63del mice is associated with proteasomes.

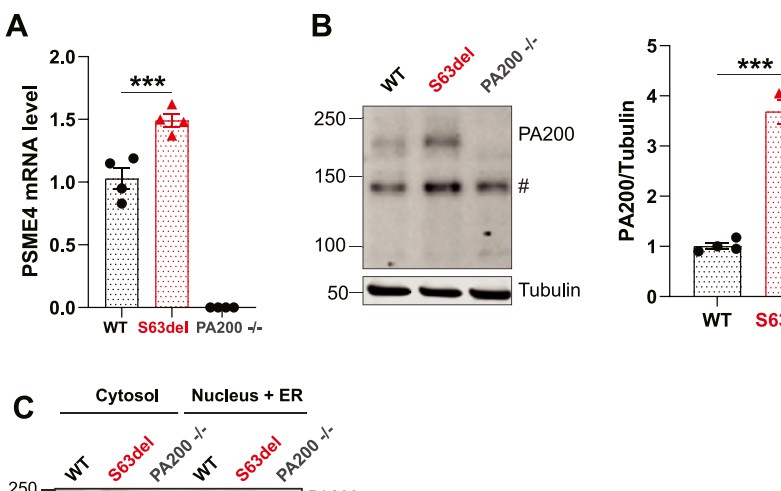

**Figure 1. PA200 mRNA and protein are increased in the sciatic nerves of S63del mice.**
**(A)** *PSME4* mRNA is increased in the sciatic nerves of S63del mice and absent in PA200−/− mice. n = 4 mice per genotype. One-way ANOVA with Bonferroni post-hoc comparison between WT and S63del. Here and throughout, error bars are SEM and * = $P$ 0.05; ** = $P$ 0.01; and *** = $P$ 0.001. **(B)** PA200 protein is increased in sciatic nerve lysates of S63del mice. # indicates a band detected in the sciatic nerve lysates of PA200−/− mice that is likely nonspecific. n = 4 mice per genotype. $t$ test. **(C)** PA200 protein is located predominantly in the cytosolic fraction of sciatic nerves in WT and S63del mice. The increased levels of PA200 protein in S63del mice is seen in both the cytosolic and nuclear fractions. The experiment was repeated twice, with samples from different mice with similar results. Representative Western blots are shown.

There are three types of PA200-bound proteasomes: a 20S with PA200 on one end (PA200-20S), a 20S with PA200 on both ends (PA200$_2$-20S), and a 20S with PA200 on one end and a 19S on the other (PA200-20S-19S) (Qian et al, 2013). To identify which types of PA200-bound proteasomes are present in the sciatic nerves and increased in S63del, we separated sciatic nerve lysates by native PAGE and performed Western blot for PA200 and the 20S proteasome subunit α4. The most prominent band detected with an antibody against PA200 that was also absent in sciatic nerve lysates from PA200−/− mice co-migrated with the 20S (Fig 2C), indicating a 20S proteasome with PA200 on one end (PA200-20S). These PA200-20S proteasomes were the most abundant type of PA200-bound proteasomes in sciatic nerves, and their levels were much greater in S63del than in WT (Fig 2C). Faint higher molecular weight bands were detected in S63del that were absent from the lysates of PA200−/− mice (Fig 2C). These bands migrated at approximately the same size as the 26S proteasome (19S-20S) and are most likely PA200$_2$-20S and PA200-20S-19S proteasomes. However, based on band intensities, these must represent a small fraction of the population of PA200-bound proteasomes in Schwann cells.

PA200-20S proteasomes were found predominantly in the cytosolic fractions of sciatic nerves from both WT and S63del mice (Fig 2D). In S63del, PA200-20S proteasomes were increased in both the cytosolic and nuclear fractions (Fig 2D), but most noticeably in the cytosolic fraction. The levels of PA200-20S proteasomes and their subcellular locations thus seem to resemble those of PA200 protein (Fig 1C).

We previously reported that the amount of assembled 26S proteasomes is increased in the sciatic nerve lysates of S63del mice (VerPlank et al, 2018), and this was also seen here in lysates (Fig 2C), and in cytosolic and nuclear fractions (Fig 2D). In nuclear fractions, 26S and 20S proteasomes were readily detected in S63del, but not in WT (Fig 2D). This finding was unexpected and suggests an increase in nuclear proteasomes in S63del. However, it should be

noted that our fractionation method does not separate the ER from the nucleus. Therefore, the proteasomes detected in the nuclear fractions generated from the sciatic nerves of S63del mice may not be in the nucleus, but instead associated with the ER; where the mutant MPZ accumulates (Sidoli et al, 2016). Whatever the reasons for this unexpected finding, in S63del sciatic nerve lysates, there is a clear increase in the amount of PA200 and PA200-bound proteasomes, in both the nuclear and cytosolic fractions.

## The knockout of PA200 in S63del mice increases protein degradation by proteasomes

We hypothesized that the increase in PA200 and PA200-bound proteasomes in the sciatic nerves of S63del mice was a compensatory response to maintain proteasomal protein degradation when there is a decrease in the 26S proteasome's ability to degrade proteins. To test this hypothesis genetically, we generated S63del mice that lacked PA200 (S63del//PA200−/−) and examined protein degradation and proteasome function in the sciatic nerves.

To measure protein degradation, we used a well-validated pulse-chase approach in which newly synthesized proteins are labeled with [³H]tyrosine in the sciatic nerves ex vivo (VerPlank et al, 2018). In sciatic nerve explant cultures, this technique primarily measures protein turnover in Schwann cells because they constitute ~90% of the dry weight of peripheral nerves (Rutkowski et al, 1992) and perform the majority of protein synthesis ex vivo because of the absence of neuronal cell bodies.

With a short pulse of 20 min with [³H]tyrosine, followed immediately by measurements of radiolabeled TCA-soluble peptides in the chase media, we first investigated the degradation of proteins which are degraded rapidly after synthesis. These proteins have short half-lives, include misfolded, regulatory (e.g., transcription factors, rate-limiting enzymes), and damaged proteins, and are degraded almost exclusively by the 26S proteasome in a ubiquitin-

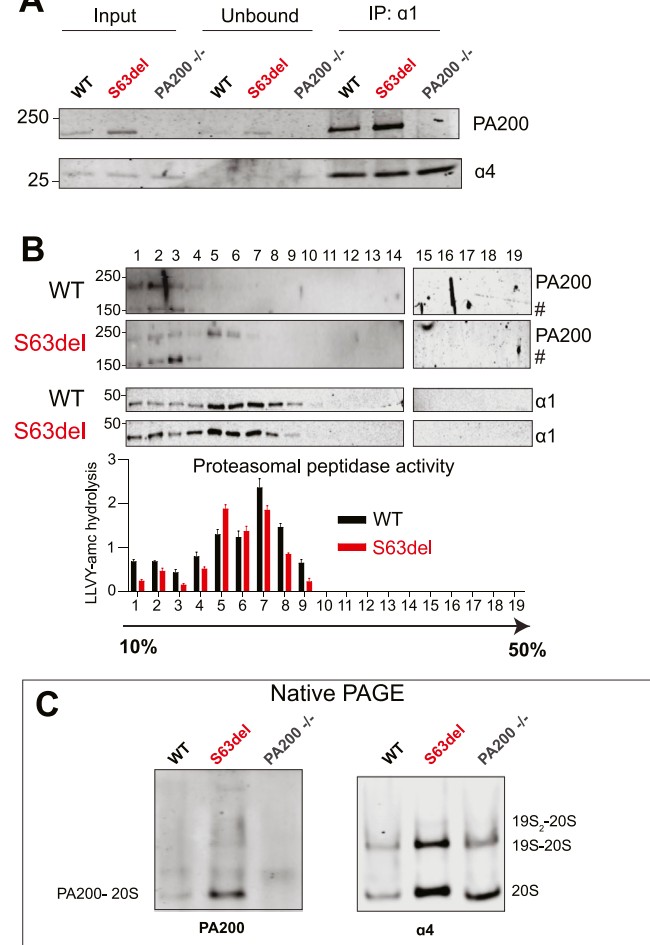

**Figure 2. PA200-bound proteasomes are increased in the sciatic nerves of S63del mice.**
**(A)** More PA200 protein co-immunoprecipitated with proteasomes in sciatic nerve lysates from S63del mice than from WT littermates. Sciatic nerves from two mice per genotype were pooled for each experiment. Representative Western blots from one of two independent immunoprecipitations are shown. **(B)** PA200 co-fractionated with proteasome subunits when sciatic nerve lysates were separated on a 10–50% glycerol gradient and PA200 was found only in fractions with proteasomal chymotrypsin-like activity. Sciatic nerves were combined from three mice per genotype and representative data from one of two independent experiments are shown. **(C)** PA200-20S proteasomes are increased in S63del sciatic nerve lysates. Representative Western blots are shown from one of three independent experiments. **(D)** PA200-20S proteasomes are found predominantly in the cytosolic fractions of WT and S63del sciatic nerve lysates and in S63del, PA200-20S proteasomes are increased in both fractions. Representative Western blots are shown from one of two independent experiments. # indicates bands detected in sciatic nerve lysates from PA200−/− mice and are likely nonspecific.

dependent manner (Sha et al, 2018b). The rate of degradation of these short-lived proteins was similar in sciatic nerves explanted from PA200−/− and WT mice, and reduced in the sciatic nerves explanted from S63del mice (Fig 3A) (VerPlank et al, 2018). In sciatic nerves from S63del//PA200−/− mice, the rate of degradation of short-lived proteins was similar to that measured in S63del (Fig 3A). Thus, in WT or S63del mice, PA200 does not participate in the degradation of these short-lived proteins, which compose a very small fraction of cell protein content.

To measure the degradation of long-lived proteins, which constitute the vast majority of cell proteins, the pulse with [$^{3}$H] tyrosine was extended to 18 h and we included a chase period of 2 h to exclude the degradation of short-lived proteins from the analysis. The sciatic nerves explanted from PA200−/− mice showed a slight (~20%), but consistent decrease in the rate of degradation of these long-lived proteins (Fig 3B and C). This finding was unexpected. PA200 has been shown to participate in the proteasomal degradation of specific proteins (e.g., acetylated histones) (Qian et al, 2013; Mandemaker et al, 2018), but this is the first evidence that it contributes to protein degradation generally. In S63del, the rate of degradation of long-lived proteins was ~50% less than that of WT (Fig 3B and C), as seen previously (VerPlank et al, 2018). Surprisingly, the sciatic nerves explanted from S63del//PA200−/− mice exhibited a rate of degradation that was ~twofold greater than that of S63del (Fig 3B and C).

Long-lived proteins, unlike short-lived proteins, are degraded by both the proteasome and the autophagy-lysosome system. To determine which was responsible for the observed changes in degradation in the absence of PA200, the assay was repeated in the presence of an inhibitor of the proteasome (Bortezomib), to measure protein degradation by the lysosome, or an inhibitor of lysosome acidification (chloroquine), to measure degradation by the proteasome (Sha et al, 2018b). Proteasome degradation was decreased by ~25% in the sciatic nerves explanted from PA200−/− mice (Fig 3D) and ~50% from S63del (Fig 3D) (VerPlank et al, 2018). In S63del//PA200−/−, the rate of proteasomal degradation of long-lived proteins was, again, ~twofold greater than that measured in S63del (Fig 3D). Protein degradation by the lysosome was similar across all four genotypes (Fig 3D). Thus, the decreased degradation of long-lived proteins in PA200−/− and S63del occurred via the proteasome, as did the increased degradation in S63del//PA200−/−.

## S63del//PA200−/− mice have greater proteasome peptidase activity and more assembled, active 26S proteasomes than S63del

Because the knockout of PA200 in WT and S63del mice altered proteasomal degradation, we assayed in sciatic nerve lysates the proteasomal hydrolysis of peptides specific to its chymotrypsin-like (LLVY-amc) and caspase-like (nLPnLD-amc) active sites. In PA200−/−, the hydrolysis of both peptides was slightly less than WT (Fig 4A and B), as would be expected with the observed decreases in proteasomal protein degradation (Fig 3). The levels of proteasome subunits in the lysates from WT and PA200−/− mice were comparable (Fig 4C). Thus, the slight decrease in proteasome peptidase activity was not because of a decreased number of proteasomes, but most likely because of the absence of PA200 and its activation of the 20S proteasome.

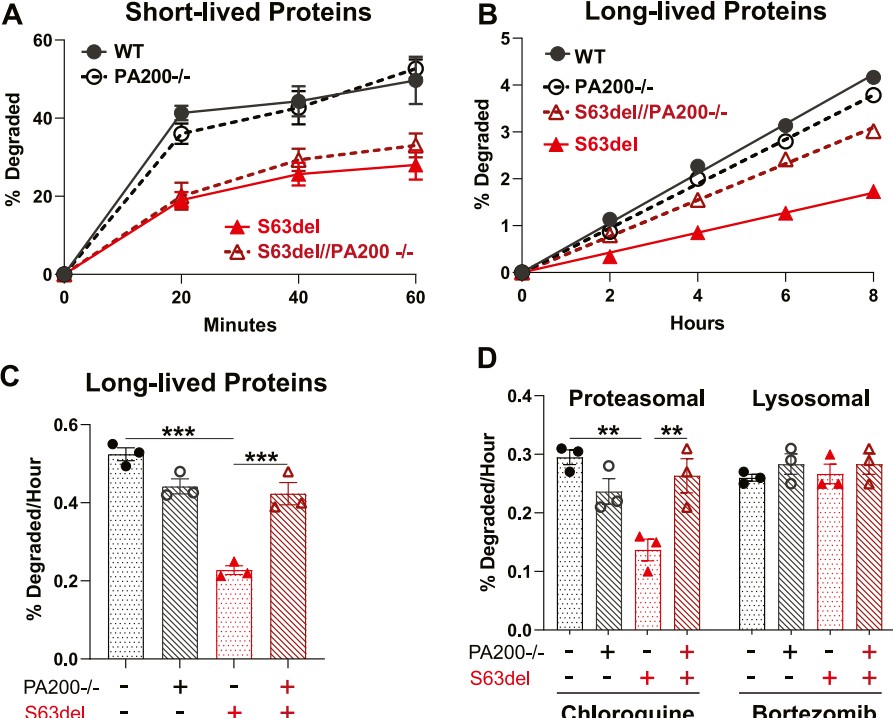

**Figure 3. Sciatic nerves from S63del//PA200−/− mice exhibit faster proteasomal protein degradation ex vivo than those from S63del mice.** **(A)** Knocking out PA200 in WT and S63del mice does not affect the degradation of short-lived proteins. The sciatic nerves of S63del mice ex vivo exhibited a slower rate of degradation of this class of proteins than WT. n = 3 mice per genotype. Experiment was repeated with similar results. **(B, C)** The degradation of long-lived proteins is slower in S63del sciatic nerves than in WT. Knocking out PA200 in S63del increased the rate of degradation of this category of proteins. Here and below, n = 3 mice per genotype and a one-way ANOVA were performed with a Bonferroni post-hoc analysis comparing WT and PA200−/−, WT and S63del, and S63del and S63del//PA200−/−. The experiment was repeated with similar results. **(D)** Protein degradation by the proteasome, measured in the presence of the inhibitor of lysosomal acidification chloroquine, is reduced in the sciatic nerves of S63del mice. S63del//PA200−/− sciatic nerves exhibit a faster rate of degradation of long-lived proteins ex vivo than those from S63del. Degradation by the lysosome, measured in the presence of the proteasome inhibitor Bortezomib, was unchanged across all four genotypes.

In S63del mice, the hydrolysis of these peptides by the proteasome in sciatic nerve lysates occurred at similar rates as in WT (Fig 4A and B) and (VerPlank et al, 2018, 2022). The lysates from S63del mice contained higher levels of proteasome subunits and assembled 26S proteasomes than those of wild-type littermates (Figs 2C and D and 4B and C) and (VerPlank et al, 2018, 2022). In S63del, the decreased degradation of proteins by the proteasome induces the upregulation of proteasome subunits (VerPlank et al, 2018), as is seen in many mammalian cultured cell lines in response to pharmacological inhibition of the proteasome (Sha & Goldberg, 2014). This increase in proteasome number is likely the reason total proteasome peptidase activity in the lysates is similar in S63del and WT, even though in S63del, the specific activity of the proteasome is decreased (VerPlank et al, 2018).

Lysates from sciatic nerves of S63del//PA200−/− mice exhibited proteasomal chymotrypsin-like and caspase-like activity that was 50% greater than that of S63del (Fig 4A and B). Thus, the knockout of PA200 in WT and S63del seems to have opposite effects on proteasome peptidase activity and protein degradation in the sciatic nerves–in WT, an expected slight decrease, and in S63del, an unexpected increase.

To better understand how the knockout of PA200 paradoxically increased proteasome activity and protein degradation in S63del mice, we first investigated the levels of 26S proteasome subunits. A greater number of 26S proteasomes in S63del//PA200−/− than in S63del could increase both protein degradation and proteasomal peptidase activity. Western blot analysis for several 26S proteasome subunits showed lower levels in S63del//PA200−/− sciatic nerve lysates than in S63del (Fig 4C), similar to those seen in WT and PA200−/− (Fig 4C). This return of proteasome subunits to WT levels

is likely because of the lessening of the proteotoxic stress and proteasome functional impairment that caused their upregulation, as seen previously when proteasomal activity and protein degradation were increased in S63del mice by a 3-wk treatment with the PDE5 inhibitor sildenafil (VerPlank et al, 2022).

Next, we evaluated the levels of assembled 26S proteasomes in the sciatic nerve lysates by native PAGE followed by Western blot analysis for the 20S subunit $\alpha$1. S63del//PA200−/− had more 26S proteasomes, especially those that are doubly capped (19S$_2$-20S), than S63del, where levels of assembled 26S proteasomes are already greater than that seen in WT (Figs 2C and 4D). An increase in assembled 26S proteasomes was also seen in the sciatic nerve lysates of PA200−/− mice (Figs 2C and D and 4D), but not to the extent of that seen in S63del//PA200−/− (Fig 4D). In both PA200−/− and S63del//PA200−/−, this increase occurred without a corresponding increase in the levels of proteasome subunits (Fig 4C), suggesting a change in stability or assembly of the 26S proteasome. Perhaps the absence of PA200 leaves ends of the 20S proteasome unoccupied and available for binding by 19S (see the Discussion section).

Not all 26S proteasomes in cells are in an active conformation that is competent to degrade proteins (Asano et al, 2015; Albert et al, 2020). To assess whether the greater number of assembled 26S proteasomes in S63del//PA200−/− sciatic nerve lysates was increasing the amount of active proteasomes, we used an irreversible inhibitor of the proteolytic active sites of the proteasome that is conjugated to the fluorescent molecule Bodipy (BodipyFL-Ahx$_3$-LeuVS) (Gan et al, 2019). Fluorescent-labeling of the proteasome active sites, which can be detected after separation of the lysates by SDS–PAGE, thus indicates active proteasomes. Sciatic nerve lysates

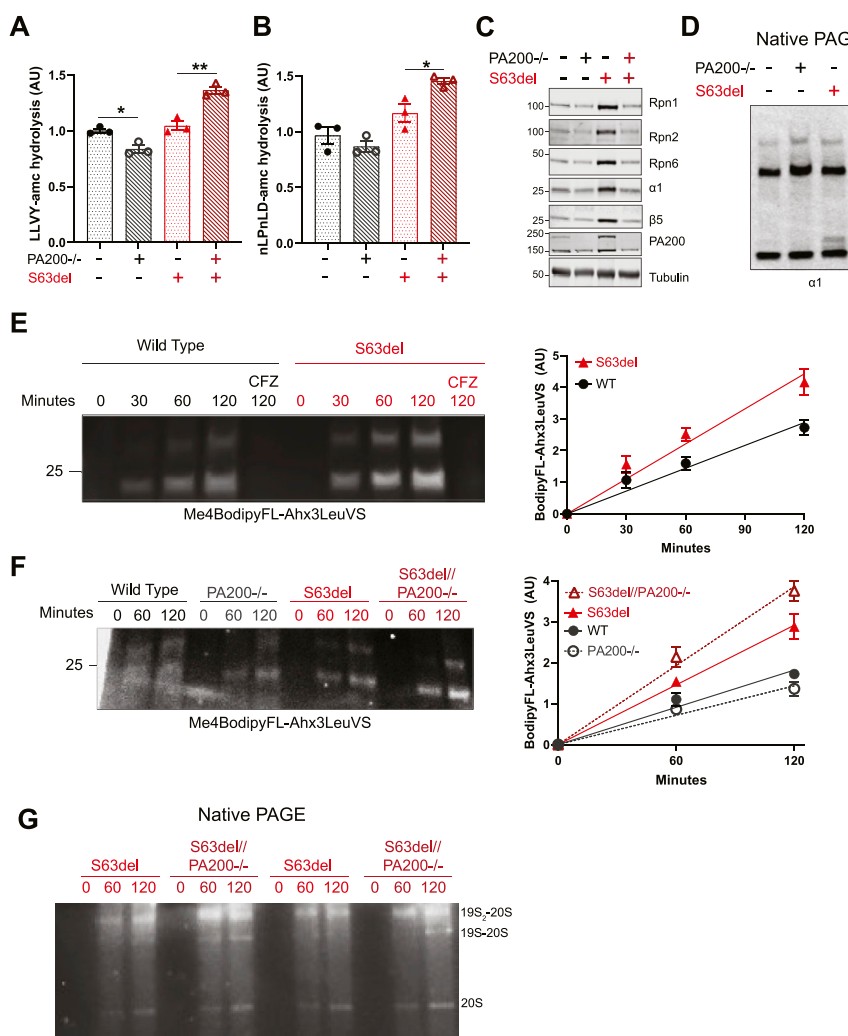

**Figure 4. Sciatic nerves from S63del//PA200−/− mice have increased proteasome peptidase activity and more assembled, active 26S proteasomes than S63del.**

**(A, B)** Proteasomal hydrolysis of peptides specific to its (A) chymotrypsin-like and (B) caspase-like active site is higher in lysates from sciatic nerves of S63del//PA200−/− mice than from S63del. n = 3 mice per genotype. Experiment was repeated with similar results. **(C)** The levels of proteasome subunits are greater in sciatic nerve lysates in S63del than in WT but return to WT levels in S63del//PA200−/−. Representative Western blots are shown of three independent experiments. n = 3 mice per genotype. **(D)** More assembled 26S proteasomes are detected in sciatic nerve lysates of S63del//PA200−/− mice than of S63del. Representative Western blots are shown of three independent experiments. **(E)** In S63del sciatic nerve lysates, there is more labeling over time of proteasome active sites with the fluorescent irreversible proteasome inhibitor Me4BodipyFL-Ahx3LeuVS than in WT. Here and in (F), representative images are shown of three independent experiments and graphs show the densitometry of the fluorescent bands. n = 3. **(F)** S63del//PA200−/− sciatic nerve lysates show more labeling of proteasome active sites with Me4BodipyFL-Ahx3LeuVS than S63del. **(G)** S63del//PA200−/− sciatic nerve lysates show more labeling of 26S proteasomes ($19S_2$-20S and 19S-20S) with Me4BodipyFL-Ahx3LeuVS than S63del. Representative image of two independent experiments.

from WT and S63del mice were incubated with BodipyFL-Ahx₃-LeuVS for 30, 60, and 120 min. Bodipy-labeling of the active proteasome subunits increased linearly over time (Fig 4E), and was blocked by co-incubation with the proteasome inhibitor Carfilzomib (Fig 4E). S63del sciatic nerve lysates showed greater labeling at all time points than WT (Fig 4E), indicating a greater amount of active proteasome in S63del. This further shows proteasome dysfunction in S63del because an increased number of active proteasomes (Fig 4E) produced a similar amount of peptide hydrolysis (Fig 4A and B) as WT, which had a lesser number of active proteasomes (Fig 4E).

Lysates of sciatic nerves from S63del//PA200−/− mice showed even greater bodipy-labeling of proteasome proteolytic subunits than S63del (Fig 4F). Analysis of the S63del and S63del//PA200−/− samples by Native PAGE showed increased bodipy-labeling in S63del//PA200−/− of 26S proteasomes and no change in labeling of 20S proteasomes (Fig 4G). Thus, S63del//PA200−/− sciatic nerves have a greater amount of assembled, active 26S proteasomes than S63del. This increased level of active proteasomes, seen in S63del//PA200−/− but not in PA200−/−, is likely responsible for the

increased proteasomal degradation of proteins and hydrolysis of peptides in the sciatic nerves of S63del//PA200−/− mice.

## S63del//PA200−/− sciatic nerves have less proteotoxic stress than S63del

The knockout of PA200 in S63del increased proteasomal peptidase activity (Fig 4) and proteasomal degradation of proteins (Fig 3) in sciatic nerves. We've shown previously that increasing proteasome function in S63del by pharmacologically raising cGMP and stimulating phosphorylation of 26S proteasomes by PKG reduces the biochemical indicators of the proteotoxicity seen in S63del neuropathy (VerPlank et al, 2022). We therefore examined whether the increased 26S proteasome activity seen here in S63del by the knockout of PA200 also reduced proteotoxicity.

Polyubiquitinated proteins accumulate in the Schwann cells in the sciatic nerves of S63del mice (Fig 5A) and (VerPlank et al, 2018) because of their reduced degradation by the 26S proteasome (Fig 3) and (VerPlank et al, 2018). In lysates of sciatic nerves of S63del//PA200−/− mice, the levels of polyubiquitinated proteins, included

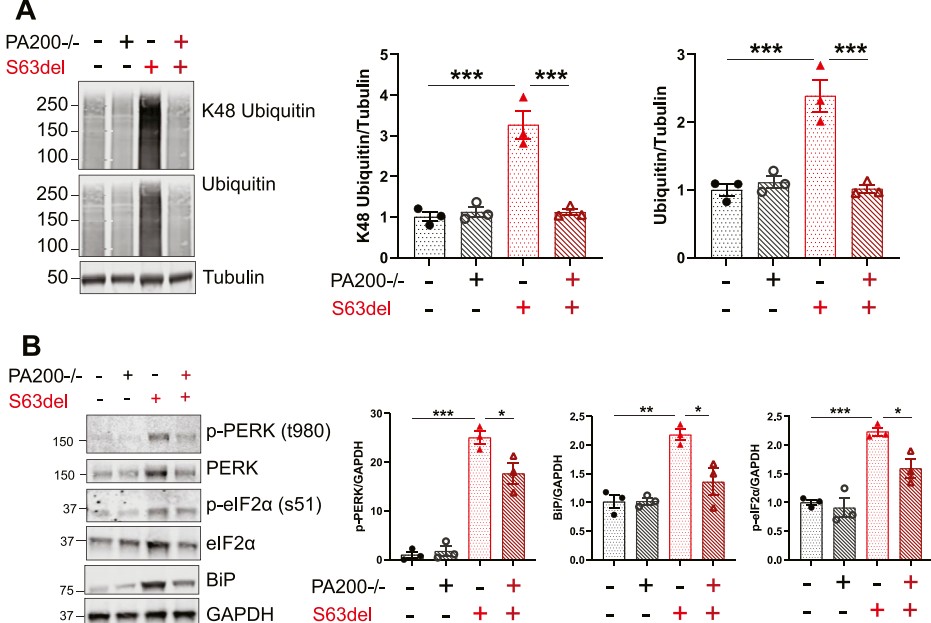

**Figure 5. Biochemical markers of proteasome impairment and proteotoxic stress are reduced in lysates of sciatic nerves of S63del// PA200−/− mice.**
**(A)** The levels of total polyubiquitinated proteins, including K48-linked polyubiquitinated proteins, were less in S63del/PA200−/− than in S63del. Representative Western blots of two independent experiments are shown. n = 3 mice per genotype. **(B)** In S63del//PA200−/− sciatic nerve lysates, the levels of PERK phosphorylated at Thr 380, eIF2α phosphorylated at Ser51, and BiP were reduced compared with S63del. Representative Western blots of three independent experiments are shown. n = 3 mice per genotype.
Source data are available for this figure.

those linked through Lysine48 which target proteins for degradation by the 26S proteasome, were less than in S63del–similar to levels seen in WT (Fig 5A). This decrease in polyubiquitinated proteins in S63del//PA200−/− is most likely because of their increased degradation by the greater number of active 26S proteasomes (Fig 4D, F, and G).

The expression of the mutant MPZ causes a canonical UPR in the Schwann cells of S63del mice (Pennuto et al, 2008) and consequently, the levels of p-PERK, p-eif2α, and BiP are greater in sciatic nerve lysates of S63del mice than in WT littermates (Fig 5B) and (Pennuto et al, 2008). In S63del//PA200−/− sciatic nerve lysates, the levels of these three markers of the UPR are decreased (Fig 5B). The reduction in S63del//PA200−/− of polyubiquitinated proteins, p-PERK, p-eif2α, and BiP indicates a lessening of the proteotoxicity in Schwann cells caused by the expression of mutant MPZ.

### The deletion of PA200 rescues the morphological and functional deficits seen in the peripheral nerves of S63del mice

S63del mice, like human patients with CMT1B, have thinner myelin sheaths around the axons of peripheral nerves (Fig 6A) (Wrabetz et al, 2006). To determine whether in S63del//PA200−/− mice the increased proteasomal protein degradation and reduced proteotoxic stress ameliorated this deficiency in myelination, the amount of myelin per axon was measured via electron microscopy of thin sections of sciatic nerves from WT, PA200−/−, S63del, and S63del// PA200−/− mice. Myelin thickness is quantified as g-ratio, which is defined as the axon diameter divided by the fiber diameter. The higher the g-ratio, the thinner the myelin sheath relative to the axon diameter. In S63del, the average g-ratio is higher than in WT (Fig 6B), indicating hypomyelination. S63del//PA200−/− mice had an average g-ratio that was lower than that of S63del and comparable to that of WT (Fig 6B). This return in myelin thickness to WT

levels in S63del//PA200−/− sciatic nerves was seen across all axon diameters (Fig 6C and D). Western blot analysis of the sciatic nerve lysates for the myelin proteins PMP22 and MBP showed less of these proteins in S63del than in WT (Fig 6E), and in S63del//PA200−/−, similar levels to WT (Fig 6E), mirroring the changes in myelin thickness (Fig 6A–D), as expected. The percentage of myelinated axons per diameter was unchanged across all four genotypes (Fig 6F). PA200−/− mice exhibited an average g-ratio and levels of myelin proteins that were comparable to that of WT (Fig 6B and E).

S63del mice model another aspect of the pathology seen in human patients with CMT1B: amyelinated fibers (red arrow in Fig 6A) (Wrabetz et al, 2006). These axons meet the requirements to be myelinated–are ensheathed by a single Schwann cell and have a diameter greater than 1 μm–but are not myelinated. Very few amyelinated fibers were present in the sciatic nerves of WT and PA200−/− mice (<1% of all myelinated axons) (Fig 6G), but more were found in S63del (~4% of all myelinated axons) (Fig 6G). The sciatic nerves of S63del//PA200−/− mice had less amyelinated fibers than S63del (~1% of all myelinated fibers) (Fig 6G).

The return of myelin thickness in S63del//PA200−/− mice to WT levels and the reduction in amyelinated fibers prompted us to investigate whether the function of the nerves was also restored. S63del mice, like patients with CMT1B, have reduced motor function because of slowing of nerve conduction velocities, prolongation of distal latencies, and prolongation of F-wave latencies that are caused by uniform hypomyelination (Fig 7A–F). In S63del//PA200−/− mice at 1 mo of age, the age at which all above analyses were performed, nerve conduction velocities (Fig 7A), distal latencies (Fig 7B), and F-wave latencies (Fig 7C) were indistinguishable from WT, indicating a restoration of the nerve conduction defects seen in S63del. At 6 mo of age, these same measures of electrical conduction in S63del//PA200−/− mice remained quicker than in S63del and comparable to WT (Fig 7D–F). Thus, the restored function in the

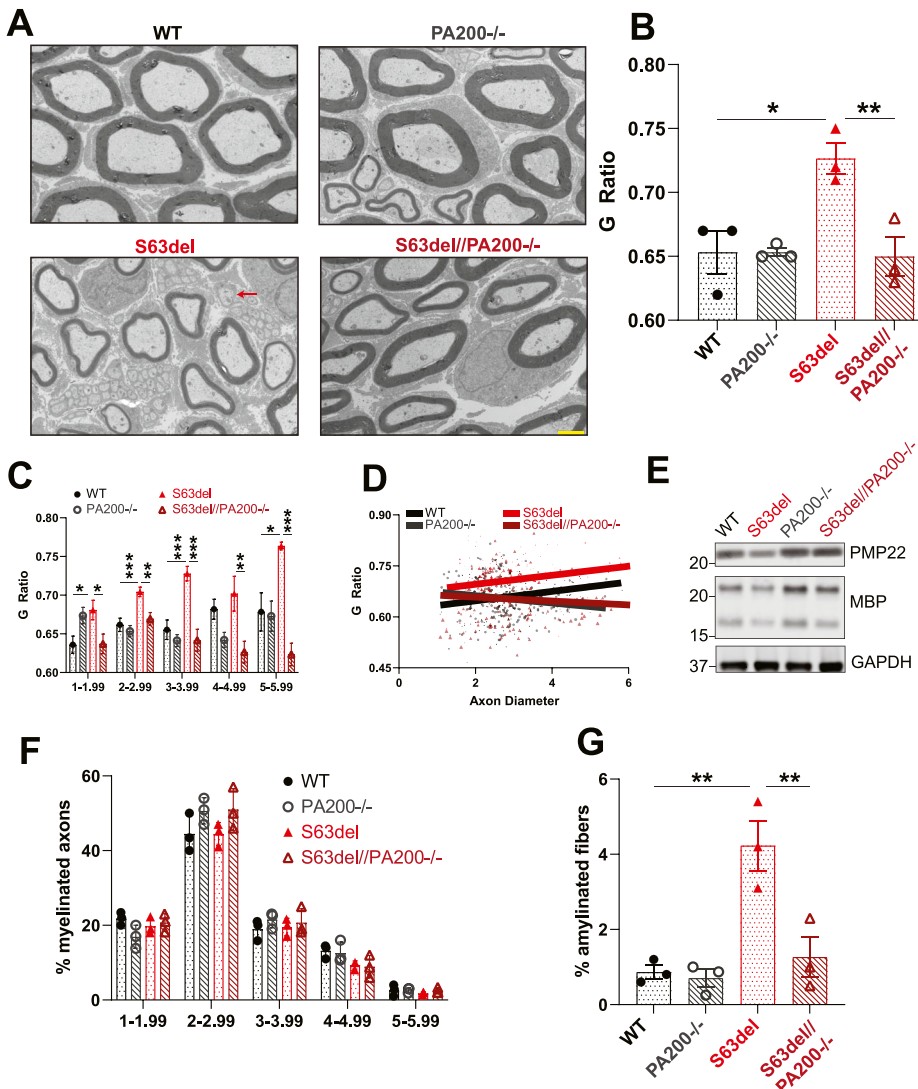

**Figure 6. S63del//PA200−/− mice lack the myelin defects seen in the sciatic nerves of S63del mice.**
**(A)** Representative electron microscopic images of thin sections of sciatic nerves showing that myelin was thicker in S63del//PA200−/− mice than in S63del mice. The red arrow indicates a pathological amyelinated fiber. Scale bar: 2 μm. **(B)** On average, the myelin surrounding all axons was thinner in S63del sciatic nerves than in WT. In S63del//PA200−/− sciatic nerves, myelin thickness was comparable to WT. g-ratio = axon diameter/fiber diameter. Therefore, the higher the g-ratio, the thinner the myelin sheath. **(C)** Myelin thickness in S63del//PA200−/− sciatic nerves is greater than in S63del across all axon diameters. **(D)** Scatterplot of the g-ratio distribution of sciatic nerve thin sections from (C). S63del//PA200−/− mice have thicker myelin than S63del (lower g-ratio) on axons of all diameters, indicated by the dark red line always being below the light red line. **(E)** Sciatic nerve lysates from S63del//PA00−/− mice have higher levels of myelin proteins PMP22 and MBP than from S63del. Representative Western blots from one of three independent experiments. n = 3 mice per genotype. **(F)** The percentage of myelinated axons of all diameters is similar across the four genotypes. **(G)** The incidence of pathological amyelinated fibers (arrow in (A)) is less in S63del//PA200−/− sciatic nerves than in S63del. n = 3 mice per genotype.

peripheral nervous system of S63del//PA200−/− mice, which is dependent on myelin thickness, is seen at 1 mo of age and maintained at 6 mo of age.

## Discussion

CMT1B is a proteotoxic disease in which there is an upregulation of proteasome subunits (VerPlank et al, 2018). Increased transcription of all subunits of the 26S proteasome is also seen when mammalian cells in culture are exposed to proteasome inhibitors (Sha & Goldberg, 2014). It is noteworthy that a chronic reduction in proteasomal protein degradation caused by the presence of a mutant protein, as seen in S63del mice, would produce a similar transcriptional response as the rapid pharmacological inhibition of the proteasome. In both instances, synthesizing more proteasomes is likely an attempt to increase degradative capacity in the affected

cells. Further studies are necessary to determine whether proteasome upregulation is also seen in other proteotoxic diseases and is therefore a general response to a chronic reduction in protein degradation by the UPS.

The transcription factor Nrf1 mediates the increased expression of 26S proteasomes genes that occurs in response to proteasome inhibitors (Sha & Goldberg, 2014). *PSME4*, the gene that encodes PA200, also has a Nrf1 binding site close to its transcriptional start site (Welk et al, 2016), and in mice in which Nrf1 was overexpressed, *PSME4* transcription was increased in livers and retina, the only two tissues tested (Wang et al, 2023). This, combined with previous data showing that proteasome subunits and PA200 are upregulated in contexts when the active form of Nrf1 is in the nucleus (Sha & Goldberg, 2014; VerPlank et al, 2018), makes it very likely that Nrf1 mediates the upregulation of *PSME4* when the proteasome is inhibited, and in S63del mice.

When degradation by proteasomes is decreased or inhibited, 26S proteasomes are upregulated, presumably to restore proteostasis by

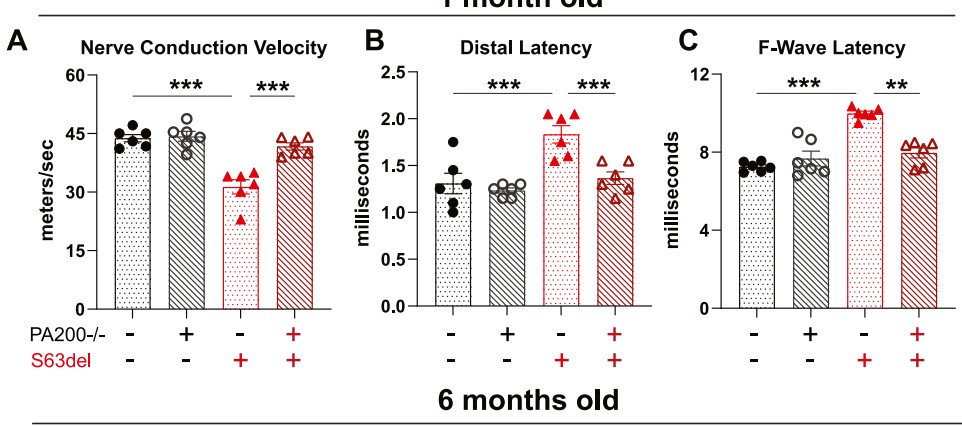

## 1 month old

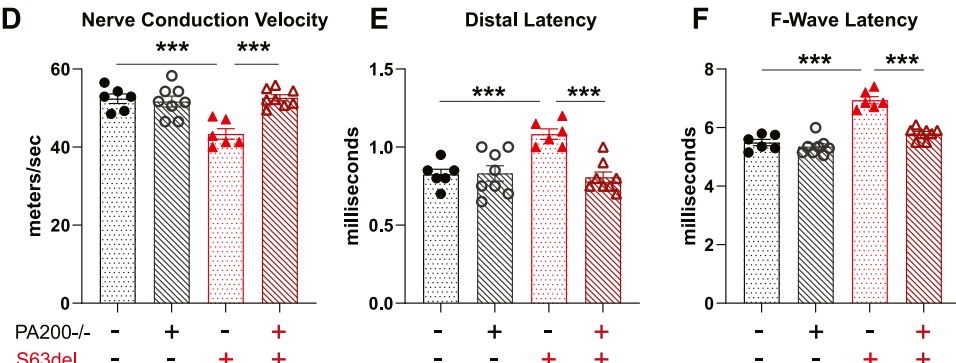

## 6 months old

**Figure 7. Knockout of PA200 in S63del mice restores measures of nerve conduction to that of wild type mice.**
**(A, B, C)** S63del//PA200−/− mice at 1 mo of age exhibit (A) faster nerve conduction velocities and (B) decreased distal latencies and (C) decreased F-wave latencies compared with S63del mice. **(D, E, F)** S63del//PA200−/− mice at 6 mo of age still exhibit (D) faster nerve conduction velocities and (E) lesser distal latencies and (F) lesser F-wave latencies than S63del mice.

increasing the degradation of the accumulated polyubiquitinated proteins. This raises the question of why also upregulate PA200? Although PA200 is an activator of the 20S proteasome–it binds the 20S alpha ring and opens the gate, permitting proteins to access the proteolytic sites within the central pore (Toste Rêgo & da Fonseca, 2019; Guan et al, 2020)—its role in intracellular protein breakdown is unclear. PA200, when it binds the 20S, sits directly atop the gate, leaving the smallest opening of all the activators through which proteins can enter the 20S for breakdown into peptides (Toste Rêgo & da Fonseca, 2019; Guan et al, 2020). PA200 also does not hydrolyze ATP and has not been shown to bind ubiquitin. Because of these constraints, PA200-20S proteasomes are unlikely to degrade polyubiquitinated proteins that must be unfolded. In our present study, we observed no decrease in the degradation of short-lived proteins in the sciatic nerves of PA200−/− mice (Fig 3A). Because the degradation of this category of proteins is ubiquitin-dependent (Sha et al, 2018b), these findings further suggest that PA200 does not participate in the proteasomal degradation of these proteins.

Purified PA200-bound proteasomes have been shown to degrade histones (Qian et al, 2013), and in human cells, the knockdown of PA200 reduces the degradation of histones (Mandemaker et al, 2018) and mutant aggregation-prone proteins that cause neurodegenerative diseases (Aladdin et al, 2020). In the sciatic nerves of PA200−/− mice, we found a small, but reproducible decrease in the degradation of long-lived proteins (Fig 3C). To our knowledge, this is the first evidence for a role of PA200 in general intracellular protein breakdown. In these experiments, it is not

possible to know which proteins are degraded less in the absence of PA200. To better understand PA200's role in intracellular protein breakdown, further studies must be conducted to identify the proteins degraded by PA200-bound proteasomes. With the recent development of mass spectrometric techniques to identify proteins undergoing degradation within the 20S proteasome (Javitt et al, 2023), this may soon be possible.

In this study, we hypothesized that the increase in PA200-bound proteasomes in S63del mice was a compensatory response to maintain some level of proteasomal protein degradation when 26S proteasomes are functionally impaired. If our hypothesis had been correct, the knockout of PA200 in S63del mice would have further reduced intracellular protein degradation by proteasomes and increased the levels of polyubiquitinated proteins and markers of the UPR. Instead, we found that the sciatic nerves of S63del//PA200−/− mice exhibited a greater rate of protein degradation than S63del (Fig 3C), and less accumulation of polyubiquitinated proteins (Fig 5A). Such a result was surprising, and we showed one unexpected mechanism through which it may have occurred: an increase of assembled, active 26S proteasomes in the sciatic nerves of S63del//PA200−/− mice (Fig 4).

Assembled 26S proteasomes were increased in the sciatic nerve lysates of both PA200−/− and S63del//PA200−/− mice (Figs 2C and D and 4D). Interestingly, there was not a concordant increase in the levels of individual proteasome subunits, as assessed by SDS–PAGE and Western blot analysis (Fig 4C). Therefore, there must be sufficient proteasome subunits free, or in subassemblies, in the

Schwann cells in sciatic nerves to assemble 26S proteasomes without an increase in expression.

It is not clear how the loss of PA200 would cause an increase in assembled 26S proteasomes. PA200 has no reported involvement in proteasome assembly, to date. The yeast orthologue of PA200, Blm10, has been reported to be involved in proteasome maturation, but yeast lacking Blm10 do not exhibit an increased amount of 26S proteasomes (Wani et al, 2015). One possible explanation for there being more assembled 26S proteasomes in S63del//PA200−/− is that, at least in Schwann cells in peripheral nerves, the absence of PA200 frees up ends of 20S proteasomes for binding by the 19S. PA200 is a single polypeptide and must only fold properly to bind 20S proteasomes. The 19S, however, consists of nineteen constitutive subunits that must be assembled in a stepwise process through several subassemblies before associating with the 20S (Mao, 2021). The PA28 complexes consist of seven subunits that must also assemble before associating with the 20S. If the 20S ends available for binding are limiting, PA200, out of the activators, could be the one most likely to associate with the 20S because it does not require assembly. Our findings are consistent with such a model, but more experiments are required to understand how one activator, and not another, binds the 20S, and how deletion of one of the nonessential activators (e.g., PA200 or PA28) influences the proteasome association of the others.

Although the sciatic nerve lysates of both PA200−/− and S63del//PA200−/− mice exhibited an increase in assembled 26S proteasomes, only S63del//PA200−/− had more proteasomal peptide hydrolysis (Fig 4A and B), protein degradation (Fig 3C and D), and active proteasomes (Fig 4F). Based on these observations, increasing the number of 26S proteasomes was not sufficient to increase proteasome peptidase activity in the lysates and protein degradation in the nerves. There must be additional regulatory mechanisms that control whether a 26S proteasome is active and competent to degrade proteins. Further study is required to identify which are present in the Schwann cells in S63del//PA200−/− mice, but not in PA200−/− mice. One possible mechanism to activate proteasomes is the binding of regulatory proteins. Prior studies in human cell lines found that the percentage of 26S proteasomes bound by the deubiquitinase USP14 correlated with the percentage of 26S proteasomes seen in active conformation in cryo-EM studies in mouse neurons (Asano et al, 2015; Kuo & Goldberg, 2017). 26S proteasomes purified from the sciatic nerves of S63del mice have more co-purifying USP14 than those from WT littermates (VerPlank et al, 2018), and more 26S proteasomes are active in sciatic nerve lysates from S63del mice than in WT (Fig 4E and F), further supporting the possible relationship between USP14 and active 26S proteasomes.

PA200 and PA200-bound proteasomes were found predominantly in the cytosolic fractions of sciatic nerves of WT and S63del mice (Fig 1C). Recently, in the human cancer cell line A549, mass spectrometric analysis showed similar levels of PA200 incorporated into proteasomes immunoprecipitated from the cytosolic and nuclear fractions (Javitt et al, 2023). PA200 has a bipartite NLS, raising the question of how a percentage of it remains localized in the cytosol. One possibility is that once PA200 is bound to cytosolic 20S proteasomes, PA200's NLS is not sufficient to localize the PA200-20S proteasomes to the nucleus. Instead, subcellular localization of the proteasome complexes may be determined by other mechanisms that have been shown to regulate proteasome import to the nucleus (de Almeida et al, 2021; Enenkel et al, 2022). In the sciatic nerve lysates of WT and S63del mice, assembled proteasomes were localized primarily in the cytosol (Fig 2D), raising the possibility that the subcellular localization of the PA200 is determined by the subcellular localization of proteasome complexes. The mechanisms that determine proteasome subcellular localization in Schwann cells, which are differentiated non-dividing cells, are not described, and require future investigation.

Activating intracellular protein degradation by the UPS is a potential approach to treat proteotoxic diseases. One promising way to increase the proteasome's ability to hydrolyze its substrates and enhance the selective degradation of some proteins in cells is phosphorylation. Pharmacological agents that raise cAMP, and thus activate PKA, stimulate phosphorylation of the 26S proteasome subunit Rpn6 (Lokireddy et al, 2015; VerPlank et al, 2019), activate the 26S proteasome, and have been shown to have therapeutic effects in vertebrate models of diseases in which the proteasome is impaired and the causative mutant protein accumulates (Myeku et al, 2016; VerPlank et al, 2019). Pharmacologically raising cGMP, and thus activating PKG, stimulates phosphorylation of a 26S proteasome component that is presently unidentified, activates the 26S proteasomes, and has therapeutic effects in zebrafish models of tauopathy (VerPlank et al, 2020) and the S63del mouse model of CMT1B (VerPlank et al, 2022).

This present study provides additional evidence that increasing degradation by 26S proteasomes is therapeutic in S63del mice. Proteasome function was increased not through pharmacological treatments that promoted phosphorylation of the proteasome, but through the surprising mechanism of knocking out a proteasome activator. Interestingly, the two approaches increased proteasomal protein degradation differently in the affected peripheral nerves of S63del mice. Raising cGMP increased the ability of pre-existing 26S proteasomes to degrade proteins (VerPlank et al, 2022), whereas knocking out PA200 in S63del seems to have increased the amount of assembled, active 26S proteasomes (Fig 4F and G). Despite these differences in how the proteasomes were activated, in both contexts there was an increase in proteasomal protein degradation in the sciatic nerves, and a restoration of proteostasis, myelin thickness, and nerve conduction.

Our data indicate that a possible new strategy to treat CMT1B neuropathy may be to prevent the formation of PA200-bound proteasomes. One approach to accomplish this could be gene silencing, for which methods are already in development for CMT neuropathies (Stavrou et al, 2021). PA200 is an attractive candidate for genetic knockdown because it is a nonessential gene and PA200−/− mice seem to have no defects aside from infertility in males (Khor et al, 2006). Furthermore, its deletion in WT mice did not negatively affect myelination or nerve conduction (Fig 7), likely because protein homeostasis in the nerves was not reduced: proteasomal degradation was not significantly altered (Fig 3) and polyubiquitinated proteins did not accumulate (Fig 5).

PA200 was recently reported to also be upregulated in non-small-cell lung carcinoma and its knockdown in a rodent lung cancer model reduced tumor burden (Javitt et al, 2023). It's noteworthy that in both CMT1B and non-small-cell lung carcinoma, two

very different diseases, the upregulation of PA200 was maladaptive. In addition, these studies highlight that the proteasome population in cells can be altered in disease states and that its modification via the genetic ablation of PA200 can have therapeutic effects.

# Materials and Methods

### Animal models

All experiments involving animals followed protocols approved by the Institutional Animal Care and Use Committee of University at Buffalo and Roswell Park Cancer Institute. Mice were housed under specific pathogen-free conditions at 70°F, 50% room humidity, 12-h light/12-h dark cycle and received ad libitum access to water and food. All mice were in a congenic C57BL/6 background and both sexes were used equally. S63del (Wrabetz et al, 2006) and PA200−/− (Khor et al, 2006) transgenic mice have been described previously. Genotyping of transgenic mice was performed by PCR using QuickLoad PCR mix (New England Biolabs) on tail genomic DNA extracted with phenol chloroform. The following three primers were used in a single reaction to detect the PA200wt allele (380 bp) and the PA200null allele (500 bp): PA200wt-AS: 5′-GTT GTT TGT TAG TTG TCA GGC TC-3′; Common-S: 5′-CCA CCA TCT AGG TTA AAG GT-3′; PA200null-AS: 5′-CCG CTC GAG GGC AGT ACA GTC TTA CT-3′. All experiments for Figs 1–6 were performed on sciatic nerves dissected from mice at ages between p28 and p32. The EMG experiments in Fig 7A–C were performed on mice at ages between p28 and p32. The EMG experiments in Fig 7D–F were performed on mice that were 6 mo old.

### RNA isolation and qRT-PCR

Nerves were first pulverized in liquid nitrogen. TRIzol (Thermo Fisher Scientific) was added (400 $\mu$l). The mixture was then passed several times through a 26-gauge needle. Chloroform (80 $\mu$l) was added, and the mixture was centrifuged at 12,000$g$ for 15 min at 4°C. The aqueous phase (180 $\mu$l) was isolated, combined with 200 $\mu$l of isopropanol, 38 $\mu$l 3M sodium acetate pH 5.5, and 1 $\mu$l of glycogen (20 mg/ml), and then incubated for 1 h at −80°C. The precipitated RNA was collected by centrifugation at 12,000$g$ for 10 min at 4°C and then washed four times by the addition of 1 ml 75% ethanol followed by centrifugation at 7,600$g$ for 5 min at 4°C. The pellets were resuspended in 10 $\mu$l DEPC water and the concentration of RNA was measured on a NanoDrop 2000c (Thermo Fisher Scientific). The RNA (0.7 $\mu$g) was reverse transcribed to cDNA using the SuperScript III First Strand Synthesis System (Invitrogen), according to the manufacturer's instructions.

TaqMan qRT-PCR was performed on a Bio-Rad CFX96/384 real-time PCR machine with the following probes: *Psme4*, Mm01193108_m1 and *GAPDH*, Hs99999915_g1. Each reaction had 10 $\mu$l of 2X Taqman Universal PCR Master Mix, 1 $\mu$l of probe, 8 $\mu$l of DEPC water, and 1 $\mu$l of cDNA. Data were analyzed using the threshold cycle (Ct) and $2^{(-\Delta\Delta Ct)}$ methods.

### Lysis for immunoblotting

Sciatic nerves were flash frozen and pulverized with mortar and pestle in liquid nitrogen. The pulverized tissues were suspended in CHAPS lysis buffer (25 mM HEPES-KOH, 150 mM NaCl, 1 mM EDTA, and 0.3% CHAPS) containing the following: 10 mM N-Ethylmaleimide, 1 mM NaF, 1 mM Na$_3$VO$_4$, 1 mM PMSF, and 1 $\mu$M Bortezomib. The crude lysates were rotated for 15 min at 4°C, sonicated (three times, 10 s on–30 s off, 25% power, Bandelin HD 2200) and centrifuged at 14,000$g$ for 10 min at 4°C. Protein in the supernatant was quantified with the detergent-compatible Bradford Assay (Life Technologies). Lysates were separated by Bis–Tris SDS–PAGE (Genscript) and transferred to either nitrocellulose (Protran; VWR) or PVDF (Immobilin FL; EMD Millipore) membranes. Immunoblotting was performed using the following antibodies per manufacturer's instructions: rb anti-ubiquitin (P4D1) (sc-8017), ms anti-PSMD1 (Rpn2) (sc-271775), ms anti-PSMB5 ($\beta$5) (sc-393931); rb anti-PSMA6 ($\alpha$1) (A303-809), rb anti-PSMD2 (Rpn1) (A303-854) (Bethyl Laboratories); rb anti-PSMA7 ($\alpha$4) (15219-1-AP), rb anti-PSMD11 (Rpn6) (14786-1-AP) (ProteinTech); ms anti-eIF2$\alpha$ (2103), rb anti-phospho eIF2$\alpha$ (3398), rb anti-K48-ubiquitin (8081), rb anti-PERK (3192), rb anti-p-PERK (3179) (Cell Signaling); rb anti-GRP78 (BiP) (NBP1-06274), rb anti-Beta-Tubulin (NB600-936) (Novus); rb anti-PA200 (PA1-1961), ms anti-LaminB1 (33–2000) (Invitrogen); rb anti-GAPDH (G9545), rb anti-PMP22 (SAB4502217) (Sigma-Aldrich); ms anti-MBP (836504) (BioLegend). When detecting ubiquitin, the nitrocellulose membranes were autoclaved after transfer. Visualization was performed with ECL, ECL Prime (GE Healthcare), or Odyssey CLx infrared imaging system (LiCor), and quantification was performed with ImageJ (National Institutes of Health) or ImageStudio (LiCor) software.

### Generation of cytosolic and nuclear fractions from sciatic nerve

Cytoplasmic and nuclear fractions were produced as previously described (VerPlank et al, 2020), with minor changes to the compositions of the buffers. Briefly, nerves were pulverized in liquid nitrogen and suspended in STM buffer (50 mM Tris–HCl pH 7.5, 250 mM sucrose, 5 mM MgCl$_2$, 1 mM ATP, 1 mM DTT, 1 mM PMSF, and 1 mM NaF). The crude lysate was centrifuged at 800$g$ for 15 min at 4°C. The supernatant (cytosolic fraction) was transferred to a new tube and the pellet (nuclear fraction) was resuspended in STM buffer, vortexed, sonicated, and centrifuged at 500$g$ for 15 min at 4°C. The pellet was washed in STM buffer, collected by centrifugation at 1,000$g$ for 10 min at 4°C, and then resuspended in NET buffer (25 mM HEPES-KOH pH 7.5, 150 mM NaCl, 5 mM MgCl$_2$, 10% glycerol, 1 mM ATP, 1 mM DTT, 0.1% NP-40, 1 mM PMSF, and 1 mM NaF). The crude nuclear fraction was incubated for 30 min on ice, sonicated, and then centrifuged at 9,000$g$ for 30 min at 4°C. The supernatant was collected as the nuclear fraction. The cytosolic fraction was centrifuged at 10,000$g$ for 10 min at 4°C to remove the mitochondria. NaCl was added to the cytosolic fractions to a final concentration of 150 mM, to be equimolar with the nuclear fractions. All fractions were normalized to the amount of total protein by detergent-compatible Bradford assay.

## Immunoprecipitation of proteasomes from sciatic nerve lysates

Sciatic nerves were flash frozen, pulverized in liquid nitrogen, suspended in APB buffer (25 mM HEPES-KOH pH 7.5, 150 mM NaCl, 5 mM $MgCl_2$, 1 mM ATP, 1 mM DTT, and 10% glycerol) plus 1 $\mu$M Bortezomib (Enzo), sonicated (three times, 10 s on–30 s off, 25% power, Bandelin HD 2200), and centrifuged at 14,000$g$ for 10 min at 4°C. Protein concentration of the clarified lysates was determined by Bradford assay and equal amounts of total protein per genotype were pre-cleared with Protein A/G Plus agarose beads (Santa Cruz Biotechnology) for 30 min at 4°C. Rabbit anti PSMA6 (Bethyl Laboratories) was added to the supernatant at a final concentration of 2 $\mu$g/ml and the samples rotated end over end for 2 h at 4°C. Protein A/G Plus agarose beads, washed 3x in APB buffer, were added and the mixture was rotated end over end for 1 h at 4°C. The beads were then collected by centrifugation and washed 3X in APB buffer for 5 min each time. The immunoprecipitates were eluted by boiling for 5 min in 2X Laemmli sample buffer (120 mM Tris–HCl pH 6.8, 4% SDS, 200 mM DTT, 0.002% xylene cyanol).

## Morphological analysis of sciatic nerves by electron microscopy

Sciatic nerves were prepared as previously described (VerPlank et al, 2022). Ultrastructural images of thin sections of sciatic nerves were acquired on a FEI Tecnai G2 Spirit BioTWIN electron microscope. These images were used for the quantification of g-ratio and axon distribution. At least 15 images and 65 axons were quantified per mouse. The analysis of amyelinated fibers was performed using Image J software on semithin images of sciatic nerves. Seven hundred and fifty fibers were counted per mouse.

## Electrophysiology

Mice were anesthetized with 2,2,2 tribromoethanol (Avertin; Sigma-Aldrich) in $H_2O$ and placed under a heating lamp to avoid hypothermia. Nerve conduction velocities, distal latencies, and F-wave latencies were obtained with steel monopolar needle electrodes. To stimulate at distal, proximal, and spinal sites along the nerve, a pair of stimulating electrodes was inserted subcutaneously near the nerve at the ankle; a second pair at the sciatic notch, and a third over the dorsum of the spine.

## Ex vivo sciatic nerve protein degradation measurements

The degradation rate of short- and long-lived proteins in sciatic nerves ex vivo was performed as described previously (VerPlank et al, 2018). Briefly, segments of equal length were dissected from the right and left sciatic nerves of each animal. To measure the degradation of long-lived proteins, the segments were incubated for 18 h at 37°C in 5% $CO_2$ in DMEM containing 10% FBS and 100 ng/ml nerve growth factor (culture media) and 5 $\mu$Ci/ml L-[3,5$^3$H]Tyrosine (Perkin Elmer). The segments were then incubated for 2 h in chase media (culture media containing 2 mM cold tyrosine) to exclude short-lived proteins from the analysis. At the end of the 2-h chase, one nerve segment per animal was transferred to fresh chase media. The other was collected and the TCA-insoluble radioactivity (i.e., radiolabeled proteins) was measured per sample on a Tri-Carb 2800TR scintillation counter (Perkin Elmer). Aliquots of media were taken at indicated times and the TCA-soluble radioactivity (i.e., radiolabeled amino acids and small peptides) was measured. Proteolysis was expressed as the amount of TCA-soluble radioactivity released over a time as a percentage of input TCA-insoluble radioactivity. To measure the degradation of long-lived proteins by the UPS or the autophagy–lysosome system, 5 $\mu$M bortezomib or 150 $\mu$M chloroquine was included in the chase media.

To measure the degradation of short-lived proteins, the following changes to the above protocol were made: nerve segments were incubated for 4 h in culture media at 37°C in 5% C02 before the pulse with 5 $\mu$Ci/ml L-[3,5$^3$H]Tyrosine, the time of pulse was reduced to 30 min, and after the pulse the nerve segments were washed 3x with chase media.

## Native gel electrophoresis

For analysis of sciatic nerve lysates in Figs 1 and 2, sciatic nerves were pulverized in liquid nitrogen, suspended in APB, lysed by sonication, and centrifuged at 14,000$g$ for 10 min at 4°C. 4% acrylamide resolving gels were prepared (90 mM Tris–HCl pH 7.5, 90 mM Boric Acid, 0.5 mM EDTA, 5 mM $MgCl_2$, 1 mM DTT, 1 mM ATP, 10% glycerol, 0.1% APS, 0.01% TEMED) and then overlaid with 2.5% acrylamide stacking gels (90 mM Tris–HCl pH 7.5, 90 mM Boric Acid, 0.5 mM EDTA, 5 mM $MgCl_2$, 1 mM ATP, 0.1% APS, 0.01% TEMED). Equal volumes of lysates that were normalized by protein concentration were added and electrophoresis was performed in a TBE-based buffer (90 mM Tris–HCl pH 7.5, 90 mM Boric Acid, 0.5 mM EDTA, 5 mM $MgCl_2$, 1 mM DTT, 1 mM ATP) at 145 V for 3–4 h at 4°C. The gels were then soaked for 15 min in a buffer pre-chilled to 4°C containing 25 mM Tris, 190 mM glycine, and 0.1% SDS and then transferred overnight at 15 V at 4°C to PVDF membranes in transfer buffer (25 mM Tris and 190 mM glycine) containing 1% MeOH.

For analysis of assembled proteasomes following subcellular fractionation, 3.5% acrylamide gels were prepared (90 mM Tris–HCl pH 7.5, 90 mM Boric Acid, 0.5 mM EDTA, 10% glycerol, 0.1% APS, 0.01% TEMED) without a stacking gel. Cytosolic and nuclear fractions were loaded into the gels, and electrophoresis was performed in a TBE-based buffer (90 mM Tris–HCl pH 7.5, 90 mM Boric Acid, 0.5 mM EDTA, 5 mM $MgCl_2$, 1 mM DTT, 1 mM ATP) at 145 V for 3–4 h at 4°C. Western analysis was performed as described above.

In Fig 4, sciatic nerves were pulverized in liquid nitrogen, suspended in APB containing 0.1% NP-40, rotated end over end for 15 min at 4°C, sonicated (three times, 10 s on—30 s off, 25% power, Bandelin HD 2200), and centrifuged at 14,000$g$ for 10 min at 4°C. Lysates were loaded into 3–8% tris-acetate gels (Thermo Fisher Scientific) and run at 145 V for 3–4 h at 4°C in a running buffer containing: 25 mM Tris, 192 mM Glycine, 5 mM MgCl2, 1 mM ATP, 0.5 mM DTT. The gels were transferred to PVDF overnight at 4°C in the following transfer buffer: 48 mM Tris, 39 mM Glycine (pH 9.0), 0.03% SDS. After transfer, the PVDF membranes were incubated in a standard transfer buffer (25 mM Tris,190 mM glycine) with 20% MeOH for 20 min and then washed in TBS before blocking.

## Measuring peptidase activity of the proteasomes in sciatic nerve lysate

Proteasome peptidase activity in sciatic nerve lysates was measured as described previously (VerPlank et al, 2018). For chymotrypsin-like peptidase activity, 20 $\mu$M Suc-LLVY-amc (Enzo) was used, and for caspase-like peptidase activity, 20 $\mu$M AC-nLPnLD-amc (Enzo) was used. Proteasome-specific cleavage was calculated by subtracting the overall rate of peptide hydrolysis in the lysates by the rate of hydrolysis measured in the presence of 5 $\mu$M Carfilzomib (Cayman Chemical Company).

## Detecting levels of active proteasomes with Me4BodipyFL-Ahx3LeuVS

Sciatic nerves were lysed as described above for analysis by native gel electrophoresis in Fig 4. The following 3X reaction mixture was prepared on ice with and without 3 $\mu$M Carfilzomib: 150 mM Tris–HCl pH 7.5, 120 mM KCl, 15 mM MgCl2, 3 mM ATP, 3 mM DTT, 0.15 mg/ml BSA, and 0.75 $\mu$M Me4BodipyFL-Ahx3LeuVS. The mixture was added to the lysates to a final concentration of 1X and the reaction took place at 37°C, shaking at 225 rpm, for the indicated amounts of time and was terminated by the addition of Laemmli buffer and boiling for 5 min. SDS–PAGE was performed, and the gels were imaged on a VersaDoc (Bio-Rad) on the FITC setting.

For analysis by native PAGE, the experiment was performed as above, but the samples were not denatured by the addition of Laemmli buffer and boiling. Instead, the samples were loaded in 3–8% tris-acetate gels and electrophoresis was performed as described above in Fig 4.

## Glycerol gradient centrifugation

20–50% glycerol gradients were prepared in a volume of 2 ml in a base buffer of 25 mM HEPES-KOH pH 7.4, 5 mM MgCl₂, 1 mM DTT, and 1 mM ATP. Sciatic nerve lysates were prepared as described above for the immunoprecipitation of proteasomes and layered on top of the gradients. Centrifugation was performed at 55,000*g* for 3 h at 4°C in a SW41i rotor in a Beckman Optima TL centrifuge. The gradients were then displaced upwards by Fluorinert and collected as 19 fractions of equal volume.

# Supplementary Information

# Acknowledgements

We thank Barry Sleckman (The University of Alabama at Birmingham) for the PA200−/− mice and Edward Hurley for his excellent technical assistance. We are grateful for funding from: NIH National Institute of Neurological Disorders and Stroke (5R01 NS045630) to ML Feltri and (R01 and R56 NS096104) to L Wrabetz.

## Author Contributions

JJS VerPlank: conceptualization, validation, investigation, visualization, and writing—original draft, review, and editing.
JM Gawron: investigation and writing—review and editing.
NJ Silvestri: investigation and writing—review and editing.
L Wrabetz: conceptualization, supervision, funding acquisition, and writing—review and editing.
ML Feltri: conceptualization, supervision, funding acquisition, and writing—review and editing.

## Conflict of Interest Statement

The authors declare that they have no conflict of interest.

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
