## [Reviewer comments · Life Science Alliance]

Life Science Alliance

Knockout of PA200 improves proteasomal degradation and myelination in a proteotoxic neuropathy

Jordan VerPlank, Joseph Gawron, Nicholas Silvestri, Lawrence Wrabetz, and Maria Feltri

DOI: <https://doi.org/10.26508/lsa.202302349>

Corresponding author(s): Jordan VerPlank, Uniformed Services University of the Health Sciences

Review Timeline:

Submission Date:	2023-08-31
Editorial Decision:	2023-10-23
Revision Received:	2024-01-17
Editorial Decision:	2024-01-26
Revision Received:	2024-01-29
Accepted:	2024-01-30

Transaction Report:

October 23, 2023

Re: Life Science Alliance manuscript #LSA-2023-02349-T

Jordan VerPlank
Uniformed Services University of the Health Sciences
4301 Jones Bridge Rd
Bethesda, Maryland 20814

Dear Dr. VerPlank,

Thank you for submitting your manuscript entitled "Knockout of PA200 improves proteasomal degradation and myelination in a proteotoxic neuropathy" to Life Science Alliance. The manuscript was assessed by expert reviewers, whose comments are appended to this letter. We invite you to submit a revised manuscript addressing the Reviewer comments.

Thank you for this interesting contribution to Life Science Alliance. We are looking forward to receiving your revised manuscript.

Sincerely,

B. MANUSCRIPT ORGANIZATION AND FORMATTING:

Reviewer #1 (Comments to the Authors (Required)):

This manuscript reports the paradoxical finding that removal of the proteasome activator PA200, stimulates protein degradation in sciatic nerves of the S63del mouse. The S63del mouse is a model for Charcot Marie Tooth 1B. Moreover, the manuscript presents data that show that the pathologic changes in sciatic nerves of the S63del mouse are reversed when PA200 is deleted. This is a very comprehensive study. The manuscript is clearly written and the experiments are well designed. In particular, the data presented in figures 6 and 7, demonstrating a reversal of the pathologies is particularly convincing. There are some aspects of the study that require the author's attention, and these are outlined below.

Figure 1C: Here the authors report the surprising finding that PA200, a nuclear protein, is predominantly cytosolic in sciatic nerve fractions. They suggest that a "non-specific" 150 kDa band recognized by some commercially available antibodies, and has a nuclear localization, could account for the reports that PA200 is an exclusively nuclear protein. This explanation is highly unlikely, since the 150 kDa band is readily resolved from PA200 on gels. Moreover, PA200 contains a bipartite nuclear localization sequence. The authors neglect to comment on this finding in the Discussion section, and should do so.

Figure 2B: This figure is confusing. The bulk of the proteasomal peptidase activity resides in fractions 5-7. The same is true for the alpha 1 subunit of the 20S proteasome. Yet most of the PA200 is found in fractions 1-3. This is at variance with the statement that the vast majority of PA200 protein is associated with the 20S proteasomes.

Figure 2D: The authors comment that their fractionation method does not separate endoplasmic reticulum from nuclei. They suggest that in S63del mice, the proteasomes found in the nuclear fraction may actually be associated with the endoplasmic reticulum and not reside in the nucleus. If this is the case, it is misleading to the reader to refer to a nuclear fraction. Perhaps it should be designated as an "ER-nuclear fraction".

Discussion: The increase in protein degradation in S63del/PA200^{-/-} mice is attributed to an increase in assembled 26S proteasomes. PA200 competes with 19S for binding to the free ends of the 20S proteasome. Absence of PA200 leads to an increase of free 20S to bind to 19S. However this model does not account for competition by PA28. Hybrid PA28-20S-19S proteasomes have been described (Cascio et. al., EMBO J. 2002, 21(11) 2636). Would deletion of PA28 in S63del mice have the same effect as deletion of PA200?

References: Some multi author references are cited with all authors, and others are cited as first author et al., The latter is recommended for this journal.

Reviewer #2 (Comments to the Authors (Required)):

The article by VerPlank and colleagues describes novel findings with regards to the involvement of the proteasome in the pathobiology of peripheral neuropathy caused by mutation in the protein Zero gene (S63del). Mice harboring this mutation model of human disorder and display a proteotoxic neuropathy. In the current study, the scientist cross-bred S63del neuropathic mice with a transgenic line deficient in Proteasome activator 200 (PA200). Unexpectedly, the absence of PA200 increased proteasomal degradation in the double transgenic mice, as compared with the neuropathic S63del mice. The authors conclude that the observed upregulation of PA200 in the S63del neuropathic mice is maladaptive and its absence prevents neuropathy. The majority of the presented studies were carefully planned and data presentation is high quality. Overall, the data supports the conclusions of the article.

Suggested improvements for the manuscript:

1) The experimental methods for the pulse-chase experiments are missing from the manuscript, which makes evaluation of the data in Fig 3 difficult. The text on page 8-9 do not entirely match the data shown. It is unclear why such distinct pulse time points were used (20 min vs 18h). It would be informative to include a representative gel image of the labeled proteins from the two pulse time points as a representation for this two very distinct pools of studied nerve proteins.

2) The method section states that " All experiments were performed on sciatic nerves dissected from mice at ages between p28 and p32", yet figure 6 includes data from 6 mo old mice.

3) Since the anti-PA200 antibody produces a non-specific band on western blots, it would be informative to include a representative panel of the genotyping data to prove the complete absence of PA200 allele in the double transgenic animals. In the PA200 $-/-$ lane, there is an additional band between PA200 and the band marked with # (Fig 1B). It is unclear where this additional PA200 antibody reactive protein resides (nucleus or cytosol) (Fig 1C).

4) The manuscript could be shortened as there are some similar comments in the introduction and the discussion. However, it would be informative to include a discussion related to the lack of a neural phenotype in the PA200 KO mice, even at 6 mo of age.

Reviewer #1 (Comments to the Authors (Required)):

This manuscript reports the paradoxical finding that removal of the proteasome activator PA200, stimulates protein degradation in sciatic nerves of the S63del mouse. The S63del mouse is a model for Charcot Marie Tooth 1B. Moreover, the manuscript presents data that show that the pathologic changes in sciatic nerves of the S63del mouse are reversed when PA200 is deleted. This is a very comprehensive study. The manuscript is clearly written and the experiments are well designed. In particular, the data presented in figures 6 and 7, demonstrating a reversal of the pathologies is particularly convincing. There are some aspects of the study that require the author's attention, and these are outlined below.

- We thank the reviewer for their kind words about our manuscript and for pointing out areas where it could be improved.

Figure 1C: Here the authors report the surprising finding that PA200, a nuclear protein, is predominantly cytosolic in sciatic nerve fractions. They suggest that a "non-specific" 150 kDa band recognized by some commercially available antibodies, and has a nuclear localization, could account for the reports that PA200 is an exclusively nuclear protein. This explanation is highly unlikely, since the 150 kDa band is readily resolved from PA200 on gels. Moreover, PA200 contains a bipartite nuclear localization sequence. The authors neglect to comment on this finding in the Discussion section, and should do so.

- We agree that comment was warranted and have added a paragraph in the Discussion section on our unexpected finding that PA200 was predominantly cytosolic in the sciatic nerve lysates. The nuclear localization of the non-specific 150 kDa band is one possible explanation for the prior reports that PA200 is localized only in the nucleus, but we chose not to comment further on this in the discussion because we feel it is more interesting to instead discuss the little-explored biology of proteasome subcellular localization. In addition to our study showing cytosolic localization of PA200, a recent report used mass spectrometric analysis to show that PA200 was equally incorporated in proteasomes immunoprecipitated from the nuclear and cytosolic fractions of a human lung cancer cell line (A549) (Javitt et al, 2023). Since PA200 has a bipartite NLS, these findings raise the question of how some percentage of PA200, which seems to vary between cell types based on these two examples, remains in the cytosol. One possibility is that once PA200 is bound to cytosolic 20S proteasomes, the NLS is not sufficient to import the whole complex into the nucleus. The 20S-PA200 proteasome would then not be regulated by PA200's NLS, but instead by the few mechanisms in mammalian cells that are known to control proteasome subcellular localization. These mechanisms have been studied primarily in rapidly dividing human cell lines. It is not yet known what controls proteasome subcellular localization in non-dividing cells, such as Schwann cells in sciatic nerves. In our study, we found that the vast majority of assembled proteasomes in sciatic nerve lysates were in the cytosolic fraction, suggesting that proteasome subcellular localization may dictate PA200 subcellular localization. We

thank the reviewer for stimulating us to think more about our experimental results and to add comments to our Discussion that could be helpful to future readers.

Figure 2B: This figure is confusing. The bulk of the proteasomal peptidase activity resides in fractions 5-7. The same is true for the alpha 1 subunit of the 20S proteasome. Yet most of the PA200 is found in fractions 1-3. This is at variance with the statement that the vast majority of PA200 protein is associated with the 20S proteasomes.

- We think we understand the source of the confusion. We did not intend to suggest that PA200 is found in the fractions with the vast majority of proteasomes and proteasome peptidase activity. The data in Figure 2B show that PA200 is found only in fractions that contain a 20S proteasome subunit and proteasome peptidase activity. PA200 is not found in fractions without proteasomes or proteasome peptidase activity - supporting our earlier immunoprecipitation data that showed that almost all PA200 co-precipitated with the proteasome. These two pieces of evidence led us to conclude that the vast majority of PA200 is proteasome-associated. We have re-written this section of the text to make that conclusion more clear.

Figure 2D: The authors comment that their fractionation method does not separate endoplasmic reticulum from nuclei. They suggest that in S63del mice, the proteasomes found in the nuclear fraction may actually be associated with the endoplasmic reticulum and not reside in the nucleus. If this is the case, it is misleading to the reader to refer to a nuclear fraction. Perhaps it should be designated as an "ER-nuclear fraction".

- We agree and have changed the labels in Figures 1C and 2D to read Nucleus + ER.

Discussion: The increase in protein degradation in S63del/PA200^{-/-} mice is attributed to an increase in assembled 26S proteasomes. PA200 competes with 19S for binding to the free ends of the 20S proteasome. Absence of PA200 leads to an increase of free 20S to bind to 19S. However this model does not account for competition by PA28. Hybrid PA28-20S-19S proteasomes have been described (Cascio et. al., EMBO J. 2002, 21(11) 2636). Would deletion of PA28 in S63del mice have the same effect as deletion of PA200?

- The reviewer brings up a very interesting point. We had attempted to examine changes in PA28-bound proteasomes in the sciatic nerve lysates of PA200^{-/-} and S63del//PA200^{-/-} mice. We had wanted to test whether the absence of PA200 increased the amount of PA28-bound proteasomes, as it had for 26S proteasomes. Had we found more PA28-bound proteasomes, it could have supported our model that PA200 outcompetes the assembly-requiring multisubunit activators for 20S free ends. Unfortunately, we could not obtain consistent results by proteasome immunoprecipitation or native PAGE, and therefore did not include the data in this manuscript. PA28-20S are more labile than 26S and PA200-20S proteasomes, which could be why we had difficulty obtaining data that was reproducible among all experiments.

- Deleting PA28 in S63del mice would certainly be an interesting experiment. 4 types of PA28 complexes, encoded by 3 genes, can be detected in cells: heteroheptamers of PA28 α and B, and homoheptamers of PA28 α , PA28B, or PA28 γ . We would predict that the double knockout of PA28 α and B, or the single knockout of PA28 γ , would not have the same positive effects in S63del mice as the deletion of PA200 because none of the three PA28 genes are upregulated in S63del, at least as measured in previously published transcriptomic data. The PA28 genes are under the transcriptional control of NF- κ B and IFN γ and there is not a strong immune response in the nerves of S63del mice. Because PA200 knockout increased protein degradation and 26S proteasome activation only in S63del mice, we think the upregulation of PA200, and the other proteasome subunits, is important to our competition model. It would be particularly interesting to knock out PA28 in a context in which it is upregulated and see if there are more 26S proteasomes. In that case, our model would predict that the levels of PA200-20S proteasomes would be unchanged because it was already outcompeting PA28, and the levels of 26S proteasomes would be increased. A good study for perhaps another day.

We added mention that PA28 is a heptamer to the section of the Discussion in which we discuss the model and added that according to our model PA200 would also outcompete PA28 because PA200 is a single polypeptide. We also included in the Discussion that further study was required to understand how deletion of one of the non-essential activators (e.g., PA200 or PA28) influences the proteasome association of the others.

References: Some multi author references are cited with all authors, and others are cited as first author et al., The latter is recommended for this journal.

- We appreciate the reviewer catching this inconsistency. We have updated the formatting of all references to the style recommended by Life Science Alliance.

Reviewer #2 (Comments to the Authors (Required)):

The article by VerPlank and colleagues describes novel findings with regards to the involvement of the proteasome in the pathobiology of peripheral neuropathy caused by mutation in the protein Zero gene (S63del). Mice harboring this mutation model of human disorder and display a proteotoxic neuropathy. In the current study, the scientist cross-bred S63del neuropathic mice with a transgenic line deficient in Proteasome activator 200 (PA200). Unexpectedly, the absence of PA200 increased proteasomal degradation in the double transgenic mice, as compared with the neuropathic S63del mice. The authors conclude that the observed upregulation of PA200 in the S63del neuropathic mice is maladaptive and its absence prevents

neuropathy. The majority of the presented studies were carefully planned and data presentation is high quality. Overall, the data supports the conclusions of the article.

- We thank the reviewer for their kind words about our work.

Suggested improvements for the manuscript:

1) The experimental methods for the pulse-chase experiments are missing from the manuscript, which makes evaluation of the data in Fig 3 difficult. The text on page 8-9 do not entirely match the data shown. It is unclear why such distinct pulse time points were used (20 min vs 18h). It would be informative to include a representative gel image of the labeled proteins from the two pulse time points as a representation for these two very distinct pools of studied nerve proteins.

- In the Material and Methods section we have now included the description of the methods used to measure the degradation of short- and long-lived proteins instead of only referring the reader to a prior publication.
- We have re-written the paragraph in question that describes the data shown in Figure 2A. It now states more clearly that sciatic nerves explanted from WT and PA200^{-/-} mice have similar rates of degradation of short-lived proteins. That the rate of degradation of short-lived proteins is slower in the sciatic nerves from S63del mice than from WT. And that S63del and S63del//PA200^{-/-} mice have similar rates of degradation of short-lived proteins.
- The lengths of the pulses were altered to selectively follow proteins with short or long half-lives. We examined the degradation of these two broad categories of proteins to learn more about PA200's role in intracellular protein degradation. Short-lived proteins are degraded almost exclusively by 26S proteasomes in a ubiquitin-dependent manner. Because PA200 does not bind ubiquitinated proteins, we hypothesized that its absence would not alter the rate of degradation of short-lived proteins. The data supported this hypothesis. WT and PA200^{-/-} sciatic nerves had similar rates of degradation of short-lived proteins, as did S63del and S63del//PA200^{-/-}.

A long pulse, we used 18 hours but published protocols have used pulses of 12-24 hours, establishes an intracellular pool of radiolabeled proteins with long half-lives. Such proteins compose the vast majority of intracellular protein content. If PA200 had a role in the breakdown of some cell proteins, of all assays, this one would be the most likely to detect it due to the assay's sensitivity and the breadth of cellular protein content that is included in the analysis.

We have added some of the above details to the descriptions of the assays in the results section to provide the reader some context as to why we assayed these two categories of proteins.

The results from these assays are not quantified by radiography from gels. Instead, the radiolabeled TCA-soluble peptides and TCA-insoluble proteins are quantified in a scintillation counter. For this reason, we are unfortunately unable to include representative gel images.

2) The method section states that " All experiments were performed on sciatic nerves dissected from mice at ages between p28 and p32", yet figure 6 includes data from 6 mo old mice.

- We regret this oversight and are grateful to the reviewer for bringing it to our attention. That portion of the methods section has been updated to the following:
 - All experiments for Figures 1-6 were performed on sciatic nerves dissected from mice at ages between p28 and p32. The EMG experiments in Figures 7A-7C were performed on mice at ages between p28 and p32. The EMG experiments in Figures 7D-7F were performed on mice that were 6 months old.

3) Since the anti-PA200 antibody produces a non-specific band on western blots, it would be informative to include a representative panel of the genotyping data to prove the complete absence of PA200 allele in the double transgenic animals. In the PA200 $-/-$ lane, there is an additional band between PA200 and the band marked with # (Fig 1B). It is unclear where this additional PA200 antibody reactive protein resides (nucleus or cytosol) (Fig 1C).

- We agree that the manuscript could be improved by further evidence that PA200 is indeed knocked out in PA200 $-/-$ mice. We've added representative genotyping data (Supplemental Figure 1) to show that the PA200wt alleles are absent in PA200 $-/-$ mice. We also re-did the qRT-PCR analysis for PSME4 (the gene that encodes PA200) and this time included sciatic nerves from PA200 $-/-$ mice. PSME4 mRNA was undetectable in the sciatic nerves of PA200 $-/-$ mice. This new data is now shown in Figure 1A.
- We believe the eagle-eyed reviewer is referring to the band that is directly below the 200 kDa band in Figure 1B that is present at a faint intensity across all lanes. Upon close examination of all our blots, we see this band is present in Figure 1C at the same faint intensity across the cytosolic and nuclear+ER fractions. Because this band is difficult to perceive in the blots in comparison to the two bands at 200 kDa and 150 kDa, and shows no changes between experiments (sciatic nerve lysates and subcellular fractions), we chose not to comment on it in the text.

4) The manuscript could be shortened as there are some similar comments in the introduction and the discussion. However, it would be informative to include a discussion related to the lack of a neural phenotype in the PA200 KO mice, even at 6 mo of age.

- We have reduced the coverage in the Discussion section of topics that were also in the introduction: proteasome impairment in S63del mice, proteasome inhibitors causing the

upregulation of proteasome subunits and PA200, and activation of proteasomes by phosphorylation.

- We have added to the discussion that the knockout of PA200 did not negatively affect myelin thickness or nerve conduction, likely because protein homeostasis was not reduced: proteasomal degradation was not significantly less in the sciatic nerves of PA200^{-/-} mice and polyubiquitinated proteins did not accumulate.

January 26, 2024

RE: Life Science Alliance Manuscript #LSA-2023-02349-TR

Dr. Jordan VerPlank
Uniformed Services University of the Health Sciences
4301 Jones Bridge Rd
Bethesda, Maryland 20814

Dear Dr. VerPlank,

Thank you for submitting your revised manuscript entitled "Knockout of PA200 improves proteasomal degradation and myelination in a proteotoxic neuropathy". We would be happy to publish your paper in Life Science Alliance pending final revisions necessary to meet our formatting guidelines.

- please be sure that the authorship listing and order is correct
- please add ORCID ID for corresponding author -- you should have received instructions on how to do so
- please add the Twitter handle of your host institute/organization as well as your own or/and one of the authors in our system
- we encourage you to revise the figure legend for figure 7 such that the figure panels are introduced in an alphabetical order
- Lawrence Wrabetz and Maria Laura Feltri are assigned activities that do not constitute authorship. Please either update the contributions in our system and in the Author Contributions section of the manuscript, or let us know if these authors need to be removed.

Figure Checks:

- please include antibody information for Rpn1, 2 and 6 in the Materials and Methods section and confirm that the sizes indicated next to those blots are correct in Figure 4C
- please provide the original blots used to make Figure 5A as Source Data

A. FINAL FILES:

B. MANUSCRIPT ORGANIZATION AND FORMATTING:

Sincerely,

Reviewer #1 (Comments to the Authors (Required)):

The authors have paid close attention to the comments of the reviewers and have made the suggested changes to the manuscript. The revised manuscript is now in a form acceptable for publication without any further changes.

January 30, 2024

RE: Life Science Alliance Manuscript #LSA-2023-02349-TRR

Dr. Jordan VerPlank
Uniformed Services University of the Health Sciences
4301 Jones Bridge Rd
Bethesda, Maryland 20814

Dear Dr. VerPlank,

Thank you for submitting your Research Article entitled "Knockout of PA200 improves proteasomal degradation and myelination in a proteotoxic neuropathy". It is a pleasure to let you know that your manuscript is now accepted for publication in Life Science Alliance. Congratulations on this interesting work.

DISTRIBUTION OF MATERIALS:

Again, congratulations on a very nice paper. I hope you found the review process to be constructive and are pleased with how the manuscript was handled editorially. We look forward to future exciting submissions from your lab.

Sincerely,
